# Longitudinal Analysis of the ASES and Constant–Murley Scores, and the Internal Rotation/Shift and Jobe Tests Following Arthroscopic Repair of Supraspinatus Lesions

**DOI:** 10.3390/jpm13091304

**Published:** 2023-08-25

**Authors:** George Fieseler, Kevin Laudner, Jakob Cornelius, Stephan Schulze, Karl-Stefan Delank, René Schwesig

**Affiliations:** 1Clinic for Hand, Trauma and Orthopedic Surgery, Sports Medicine, Clinic Hann, Münden, 34346 Hannoversch Münden, Germany; 2Department of Health Sciences, Hybl Sports Medicine and Performance Center, University of Colorado, Colorado Springs, CO 80918, USA; 3Department of Orthopedic and Trauma Surgery, Martin-Luther-University Halle-Wittenberg, 06120 Halle, Germany

**Keywords:** clinical test, functional abilities, orthopedic exam, shoulder, supraspinatus, rotator cuff

## Abstract

It is essential to investigate patients post-surgery using functional surveys like the American Shoulder and Elbow Surgeons Shoulder (ASES) and the Constant–Murley shoulder (CMS) scores, as well as clinical tests, such as the Internal Rotation and Shift (IRO/Shift) and Jobe tests. In this study, 51 out of an initial 87 patients underwent an arthroscopic supraspinatus repair (22 single-row, 16 double-row, 13 debridement). Testing occurred pre-surgery, and 3 and 6 months post-surgery. Both surveys showed significant improvements over time among all 87 patients, but there were no differences between groups (lesion/no lesion) (*p* > 0.815) or time × group (*p* > 0.895). The IRO/Shift test showed a stronger ability to distinguish between both groups (positive vs. negative) with respect to the ASES and CMS scores over time, but the Jobe test did not (*p* > 0.100). Improvements in the CMS scores and the Jobe test were lower following repair compared to the ASES and IRO/Shift test. Most patients returned to adequate levels of functional abilities at 6 months post-surgery. The time required to return to activities of daily living and negative clinical tests was longer for the double-row repair patients compared to the single-row and debridement groups. In conclusion, both the functional surveys and the clinical tests demonstrated improvements following surgery.

## 1. Introduction

Rotator cuff lesions are one of the most common shoulder pathologies [1,2], with more than half of the population experiencing a lesion by the time they are 60 years old [3]. These pathologies often cause pain, decreased mobility, and decreased function during activities of daily living, resulting in the need for surgical intervention [4]. Between 1995–2009, there was a 238% increase in the number of rotator cuff repairs reported in New York alone [5]. With hundreds of thousands of these surgical interventions conducted every year [6], it is critical to understand the functional outcomes following surgery. 

One of the most common repairs is to the superior rotator cuff, or more specifically, the supraspinatus tendon. After achieving the goals of pain relief and increased functional ability, the patient ultimately returns to their normal activities of daily living. Surgeons often use anchors to restore the tendon footprint, suturing the tendon directly to the greater tuberosity [7]. Two of these techniques are known as single-row repair and double-row repair. Both techniques are commonly used, but there are conflicting results regarding their benefits and functional outcomes [8,9,10,11,12].

Although various tests are used to determine the progression of patients following a rotator cuff repair, the American Shoulder and Elbow Surgeons Shoulder (ASES) [13,14,15] and Constant–Murley shoulder (CMS) [16] scores are common patient self-evaluations used to determine functional ability. Malavolta et al. [17] previously reported that the ASES is a good prognostic tool for use at the 6-, 12-, and 24-month postoperative periods after supraspinatus repair. Similarly, the CMS has been reported to correlate with the isokinetic evaluation of the external rotation strength at the 6- and 12-month post-operative periods for rotator cuff repair [18]. In addition to functional surveys, clinical tests, such as the Jobe test [19,20] and the newly developed internal rotation and shift test (IRO/Shift test) [21,22] are used to determine the structural integrity and the functional properties of the supraspinatus tendon. Fieseler et al. [21] originally reported that the IRO/Shift test is a reliable and valid tool for assessing supraspinatus tears, demonstrating both strong intra-rater and inter-rate reliability. These authors then performed a subsequent study comparing the IRO/Shift test to the Jobe test and found that these clinical tools are comparable in the detection of supraspinatus lesions [22]. More specifically, the IRO/Shift test had a sensitivity of 96%, a specificity of 50%, and an accuracy of 77% compared to the Jobe test, which had a sensitivity of 89%, a specificity of 60%, and a similar accuracy of 77%. Unfortunately, there is little research describing the use of these questionnaires and special tests following an arthroscopic repair.

Randelli et al. [6] noted that with all the advances in rotator cuff repair techniques, it is essential for future research to investigate patient-assessed comfort and function following a surgical repair. Therefore, the primary purpose of this study was to determine the effectiveness of two popular rotator cuff patient questionnaires and two special tests following a tendon repair (double-row, single-row, debridement). Secondarily, this study used functional questionnaires and special tests to determine the effectiveness and length of healing necessary following a rotator cuff repair. More specifically, this study examined the use of the ASES, CMS, IRO/Shift test, and the Jobe test. These tools were used prior to the surgical repair of the supraspinatus tendon and again at the 3- and 6-month post-operative follow-ups to determine patient progression.

## 2. Materials and Methods

### 2.1. Subjects

Eighty-seven (36 females, 51 males; age: 53.6 ± 12.5 years; height: 1.76 ± 0.09 m; mass: 83.0 ± 15.3 kg, body mass index: 26.7 ± 3.51 kg/m^2^) patients volunteered to participate in this study. The inclusion criteria included participants ≥ 18 years who had primarily been treated conservatively by their General Physicians and Physiotherapists between 2018–2019 to restrict their shoulder pain. The participants then completed a conservative rehabilitation program which was unsuccessful. Following the failed conservative rehabilitation, magnetic resonance imaging was used to determine a possible structural lesion, and the patient was referred to a specialized shoulder physician for consideration of surgical intervention. The exclusion criteria consisted of patients who had a history of arthritis, arthrosis, or fracture. No patients had rheumatic disease; however, information pertaining to patient history of smoking, diabetes mellitus, and thyroid disease was not collected. All subjects provided informed consent prior to any data collection. This study was approved by the Martin-Luther University Halle-Wittenberg Ethics Committee (reference number: 2018-05).

Among the 87 patients who originally participated in this study, 36 were found to not have a rotator cuff tear upon arthroscopic surgery. Therefore, 51 total patients were radiologically and subsequently arthroscopically diagnosed with a supraspinatus lesion. Among these 51 patients with confirmed supraspinatus tears (17 traumatic, 34 degenerative) requiring surgery, 22 received a single-row repair, 16 received a double-row repair, and 13 received a debridement (Figure 1). The 36 patients who did not have a confirmed rotator cuff lesion were diagnosed with Bankart lesions (4), biceps tendinitis with SLAP tears (12) or biceps tendinitis with pulley-lesions (2), bursitis with subacromial impingement (11), sub-coracoid impingement (1), exophytic acromioclavicular arthrosis (4), adhesive capsulitis (1), and humeral head chondrolysis (1).

### 2.2. Procedures

This longitudinal study prospectively followed subjects from the time of their initial clinical evaluation with the specialized shoulder surgeon (pre-operative) to a 6-month post-operative follow-up evaluation. The initial pre-operative evaluation was performed by two physicians with extensive experience working with shoulder pathologies. During all three clinical evaluations (Exam 1: pre-surgery, Exam 2: 3-month post-operative, Exam 3: 6-month post-operative), the physicians used both the IRO/Shift test and the Jobe test to assess the function of the superior rotator cuff. Patients also completed the ASES and the CMS evaluations to determine their functional abilities. 

The arthroscopic repairs consisted of tendon fixation using either a single-row or double-row suture anchor, or debridement, and were performed by the same physician who had over 20 years of shoulder surgery experience. The decision to perform either a single-row or double-row repair was made intraoperatively prior to the tendon reconstruction based on the tear size, morphology, and amount of tendon retraction. Small tears were defined when no retraction or shallow “U-shape” conditions were found. In these cases, a single-row technique with two sutures and one anchor was used. Medium-sized tears, those with minor retraction but an “L-shaped” rupture, were fixed with a double-row technique using a double suture armed with a metal anchor as the medial line, arthroscopic knots, and a bioresorbable, loaded anchor device with no suture for the lateral line of the reconstruction. Large tears with higher retraction, and deep “U-shaped” or “L-shaped” conditions were closed via a double-row technique as previously described. All anchors were manufactured by Arthrex (Arthrex Company, Naples, FL, USA). Two FiberWire^®^ sutures were implanted and combined with bioresorbable anchors (SwiveLock^®^ SP) for the single-row, or for the lateral line in the double-row fixations. For the medial line of the double-row fixations, two-suture loaded metal (titanium), self-cutting anchors (CorkSrew FT) were used [23]. All surgical interventions included complete reconstruction of the tendon tear regardless of the size, retraction or morphology. There were no patients treated with a salvage procedure such as a “margin convergence”. Following surgery, all patients were prescribed a standard rehabilitation protocol, based on the extent of tendon damage and subsequent patient-specific physical characteristics and signs and symptoms, aimed at protecting the tendon repair, promoting healing, and ultimately returning the patient to their pre-injury level of physical abilities and function. 

The IRO/Shift test required the patient to stand in a relaxed position while they actively adducted and internally rotated their shoulder, moving their involved side arm behind their back and then superiorly elevating their hand along their spine until their end ROM. At this end point, the physician then applied additional passive motion in the same direction. If the patient experienced pain, then the physician followed this test with an additional special test (e.g., O’Brien’s test) to rule out labral or long head biceps tendon involvement. If the patient had pain during the adduction and internal rotation movement, but did not have pain indicating labral pathology, then the IRO/Shift test was considered positive for supraspinatus damage or load. Furthermore, none of the supraspinatus tear patients demonstrated signs of adhesive capsulitis at any point during the study. The Jobe test was also conducted standing while the physician passively elevated the patient’s involved shoulder to 90° of abduction with internal rotation. In this position, the physician then applied downward pressure while the patient attempted to resist this force. A Jobe test was considered positive in the presence of pain or abnormal weakness. Both tests were conducted bilaterally. 

The ASES questionnaire uses a 100-point scale to determine the patient’s self-reported level of pain and ability to complete activities of daily living (each worth 50 points). Scores range from 0 (worst possible shoulder condition) to 100 (best possible shoulder condition). The ASES has a reported intra-rater reliability of 0.84–0.96 [13,14,15,24] and is endorsed by the American Shoulder and Elbow Surgeons Research Committee. The Constant–Murley shoulder (CMS) score uses a scale to determine the patient’s level of pain and activities of daily living similar to the ASES questionnaire, but also objectively assesses the patient’s ROM and strength. The pain and activities of daily living section are worth a total of 35 points, while the ROM and strength are worth 65 points, for a total of 100 points. Similar to the ASES score, more points on the CMS score indicate better shoulder function. The Constant tool has a reported intra-rater reliability of 0.80–0.96 [25], and has been endorsed by the European Society of Shoulder and Elbow Surgery [25,26].

### 2.3. Statistical Analysis

All statistical analyses were performed using SPSS version 28.0 for Windows (IBM, Armonk, NY, USA). A Pearson Chi^2^-test was used to compare the categorical variables (IRO/Shift test and Jobe test) over the three time periods (pre-op, 3 months post-op, and 6 months post-op) and to detect any relationships. The longitudinal mean differences of the scores (ASES, CMS) were tested using a two-factor (time and tests: IRO/Shift test, Jobe test or type of surgical repair: single-row, double-row, debridement) univariate general linear model [27]. The differences between the means were considered statistically significant if the *p*-values were <0.05 and the partial eta-squared (η_p_^2^) values were >0.15 [28].

Based on the intraclass correlation coefficient (ICC) and the standard error of the mean (SEM) [29] calculation, the minimal detectable change was reported at a 95% (MDC_95_) confidence interval [30] for ASES and CMS.

## 3. Results

Among the 87 patients, regardless of whether a rotator cuff tear was present or not, both the ASES and CMS scores showed significant time effects, but no significant differences between groups (lesion vs. no lesion) (*p* > 0.815) or time × group (*p* > 0.895) (Table 1).

When comparing the special tests versus the survey scores, the IRO/Shift test showed a stronger ability to distinguish between both groups (positive test vs. negative test) with respect to ASES (Table 2) and CMS (Table 3) scores over time. At follow-up exams 2 and 3, the mean differences between the positive and negative IRO/Shift tests were significant (*p* < 0.001). In contrast, the Jobe test was not able to demonstrate significant differences in positive and negative tests at any time period (*p* > 0.100). However, the number of subjects in the positive and negative test groups were vastly different between exams 1 and 3. This may have affected the variance analysis.

Among the 51 patients with arthroscopically diagnosed supraspinatus tears, these patients’ ASES (*p* < 0.001) and CMS (*p* < 0.001) scores showed significant improvements over time (Table 4). These improvements were similar during both time periods (exam 1 vs. exam 2, and exam 2 vs. exam 3). The average preoperative ASES and CMS scores for these patients were 45.4 ± 16.4 and 41.6 ± 17.5, respectively. However, these scores did not increase to above 70 until the 6-month post-operative follow-up, at which time the ASES and CMS scores were 81.1 ± 16.3 and 77.5 ± 18.6. Similarly, the clinical tests did not return to predominantly negative findings until the 6-month follow-up (IRO/Shift: 90%; Jobe: 65%) (Table 5). However, the proportion of positive IRO/Shift tests and Jobe tests sharply decreased during the observed periods. This reduction in positive tests was stronger for the IRO/Shift test (from 94% to 10%, an 84% difference) than for the Jobe test (from 94% to 35%, a 59% difference). This equates to a mean difference between exam 1 and exam 3 of 82% for the IRO/Shift test and 59% for the Jobe test.

When comparing the functional outcomes (ASES and CMS) of a single-row repair, double-row repair, and debridement, all surgical techniques led to significant time effects in all periods of the recovery process (Table 6). The time effects were slightly larger for single-row repair (ASES: η_p_^2^ = 0.72; CMS: η_p_^2^ = 0.76) than for double-row repair (ASES: η_p_^2^ = 0.62; CMS: η_p_^2^ = 0.55) and debridement (ASES: η_p_^2^ = 0.65; CMS: η_p_^2^ = 0.73). The largest partial time effect for the ASES scores was observed for the debridement procedure between exams 2 and 3 (η_p_^2^ = 0.79). Similarly, debridement also had the largest partial time effect for the CMS scores between exams 2 and 3 (η_p_^2^ = 0.70). Table 7 depicts the number of positive and negative clinical tests (IRO/Shift, Jobe) over time for each surgical procedure. 

## 4. Discussion

With an increasing number of rotator cuff repairs performed annually [5], it is imperative for clinicians to understand the length of time necessary for patients to return to their activities of daily living and good tendon integrity. The results of this study demonstrate that arthroscopic repair of the supraspinatus requires approximately 6 months before functional abilities and the clinical integrity of the tendon return to adequate levels. Improvements in CMS scores and Jobe test results were slightly lower than ASES scores and the IRO/Shift test. 

Both the ASES and CMS scores showed significant improvements over time, but did not distinguish between symptomatic patients with and without a rotator cuff tear. This suggests that the ASES and CMS are useful tools for demonstrating functional impairment of the affected shoulder, and not a specific pathology. These improvements in scores also demonstrate that patients with supraspinatus tendon tears, as well as other soft tissue pathologies, can have successful functional outcomes following an appropriate surgical intervention. In the supraspinatus tear patients, both scores increased drastically from exam 1 to exam 3 (57% ASES, 60% CMS (Table 4), reflecting the time necessary for tendon healing in conjunction with a rehabilitation program. Similar results were observed with the clinical tests, with the IRO/Shift test returning to negative in 90% of the supraspinatus tear patients at 6 months, and in 65% with the Jobe test (Table 5). Improvements in the CMS scores and the Jobe test were lower following surgery compared to the ASES and IRO/Shift test. This may be because the CMS and the Jobe test require an active abduction contraction against resistance, which may have been more difficult for patients earlier in the rehabilitation process (3 months) compared to after the tendon had more time to heal (6 months). As such, patients who present with a positive Jobe test and a low CMS score 3 months post-operatively may have deficient tendon strength as opposed to tendon failure, assuming the IRO/Shift test is negative. 

Haunschild et al. [31] reported that following an arthroscopic rotator cuff repair, most patients returned to their jobs approximately 8 months post-surgery. The results of the current study support this previous work and note that the 51 patients who underwent supraspinatus repair required 6 months to return to adequate levels of function and negative clinical tests. We chose a common cutoff score of 70 to represent adequate shoulder function. At 3 months, the patients with supraspinatus repairs in this study had an ASES score of 65 and a CMS of 62. This CMS score is the same finding as a study conducted by Kukkonen et al. [32] who reported a CMS score of 62 at 3 months after rotator cuff surgery. Although this previous study did not assess CMS scores at 6 months, their 12-month score was 76, which is comparable to the current study’s 6-month score of 78. Similarly, Nabergoj et al. [33] reported their patients with supraspinatus tears had an average CMS score of 80 at 12 months post-surgery. These previous findings, in combination with the results of the current study (Table 8), suggest that CMS scores may not increase much between 6 and 12 months postoperatively; however, further research is necessary to confirm this.

The ASES and CMS scores in the patients who had double-row repairs were slightly lower than the patients with single-row repairs and debridement (Table 6). Similarly, the number of positive IRO/Shift and Jobe tests was higher among the double-row procedures at almost every time period compared to the single-row and debridement procedures (Table 7). Only the Jobe test performed 6 months post-operatively had a higher positive rate for the single-row procedure. These findings should be considered when planning rehabilitation and return to activity protocols for single-row repair, double-row repair, and debridement patients. 

There are a few limitations of this study’s methodology worth noting. The main limitation of these data is the absence of objective radiological data to demonstrate tendon healing during the rehabilitation process and to objectively support the results of this study. The results of this study provided insight into the outcomes of tendon repair 3 and 6 months post-operatively, but future studies should also assess outcomes over longer periods, including over multiple years. As previously mentioned, caution should be used when interpreting the results of the comparisons between the special tests and the survey scores because the number of subjects in the positive and negative special test groups were vastly different between exams 1 and 3, which may have affected the variance analysis. Psychological distress may have also played a factor in the patients’ functional outcomes. A previous systematic review investigated the association between psychosocial factors and patient-reported outcomes among rotator cuff repair patients [34]. This review found that negative psychosocial factors were associated with decreased function and increased pain both pre- and post-operatively. While the ASES may perform better psychometrically than the Patient-Reported Outcome Measurement Information System (PROMIS) [35], the impact of patient psychometric properties on tests such as the CMS, IRO/Shift test, and the Jobe test are still unclear. Finally, although all subjects with repaired tendons were prescribed the same rehabilitation protocol, not all subjects progressed through their respective protocol at the same pace, and the authors could not ensure each patient performed the protocol precisely. 

## 5. Conclusions

The ASES and CMS scores, as well as the IRO/Shift and Jobe clinical tests showed significant improvements over time following supraspinatus repair. The main results and suggestions are:The IRO/Shift test showed a stronger ability to distinguish between those with positive tests compared to negative tests with respect to ASES and CMS scores.The average for all tools did not improve to acceptable levels until approximately 6 months post-surgery, with improvements in the CMS scores and Jobe test being slower at all time periods compared to the ASES and IRO/Shift test.The time required to return to activities of daily living and negative clinical tests was longer for the double-row repair patients compared to single-row and debridement.These results suggest that the ASES, CMS, IRO/Shift test, and Jobe test are useful, yet unique tools for determining the progress and clinical decision-making process among patients with supraspinatus repair.

## Figures and Tables

**Figure 1 jpm-13-01304-f001:**
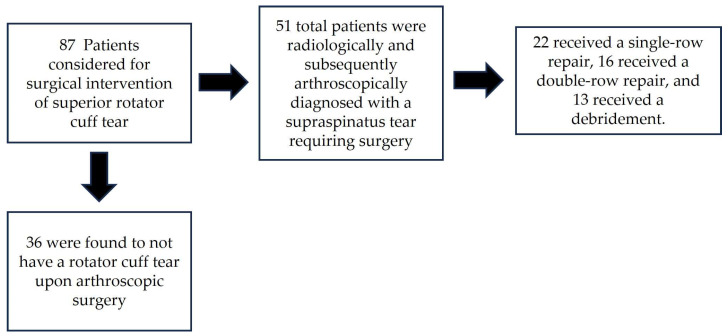
Participant flow diagram.

**Table 1 jpm-13-01304-t001:** Analysis of ASES and CMS scores dependent on the diagnosis of a superior rotator cuff lesion.

Score	Exam	RC-Lesion (n = 51)	No RC-Lesion (n = 36)	Variance Analysis (*p*/η_p_^2^)		
				Time	Group	Time × Group
**ASES**	**1**	**45.4 ± 16.4**	44.2 ± 18.4	**<0.001/0.653**	0.828/0.001	0.913/0.001
	2	65.0 ± 16.4	65.2 ± 15.6			
	3	81.1 ± 16.3	80.3 ± 16.7			
**CMS**	1	41.6 ± 17.5	42.4 ± 18.5	**<0.001/0.617**	0.815/0.001	0.895/0.001
	2	61.9 ± 17.7	61.6 ± 15.6			
	3	77.5 ± 18.6	79.0 ± 21.6			

ASES = American Shoulder and Elbow Surgeons Score; CMS = Constant–Murley Shoulder Score; RC = rotator cuff; Exam 1 = pre-operative; Exam 2 = 3 months post-operative; Exam 3 = 6 months post-operative. Significant differences marked in bold.

**Table 2 jpm-13-01304-t002:** Comparison of ASES scores to clinical tests.

Clinical Test		Exam 1 ASES Score	Exam 2 ASES Score	Exam 3 ASES Score
**IRO/Shift**	negative	50.1 ± 20.4 (n = 25)	69.7 ± 15.3 (n = 50)	82.6 ± 14.8 (n = 81)
	positive	42.8 ± 15.3 (n = 62)	58.8 ± 14.8 (n = 37)	55.8 ± 17.2 (n = 6)
*p*/η_p_^2^		*0.072/0.04*	** *0.001/0.12* **	** *<0.001/0.18* **
**Jobe**	negative	45.2 ± 19.4 (n = 31)	67.8 ± 14.6 (n = 45)	82.0 ± 16.3 (n = 67)
	positive	44.8 ± 16.0 (n = 56)	62.2 ± 16.9 (n = 42)	76.6 ± 16.2 (n = 20)
*p*/η_p_^2^		*0.902/0.00*	*0.100/0.03*	*0.193/0.02*

ASES = American Shoulder and Elbow Surgeons Score; IRO/Shift = internal rotation and shift; Exam 1 = pre-operative; Exam 2 = 3 months post-operative; Exam 3 = 6 months post-operative. Significant differences marked in bold.

**Table 3 jpm-13-01304-t003:** Comparison of Constant–Murley scores to clinical tests.

Clinical Test		Exam 1 CMS Score	Exam 2 CMS Score	Exam 3 CMS Score
**IRO/Shift**	negative	45.2 ± 20.6 (n = 25)	67.6 ± 16.2 (n = 50)	80.3 ± 16.9 (n = 81)
	positive	40.6 ± 16.5 (n = 62)	54.0 ± 14.4 (n = 37)	45.0 ± 21.8 (n = 6)
*p*/η_p_^2^		*0.279/0.01*	** *<0.001/0.16* **	** *<0.001/0.22* **
**Jobe**	negative	42.1 ± 18.6 (n = 31)	64.5 ± 16.9 (n = 45)	79.3 ± 19.9 (n = 67)
	positive	41.8 ± 17.5 (n = 56)	58.9 ± 16.4 (n = 42)	72.8 ± 16.6 (n = 20)
*p*/η_p_^2^		*0.958/0.00*	*0.125/0.03*	*0.188/0.20*

CMS = Constant–Murley Shoulder Score; IRO/Shift = internal rotation and shift; Exam 1 = pre-operative; Exam 2 = 3 months post-operative; Exam 3 = 6 months post-operative. Significant differences marked in bold.

**Table 4 jpm-13-01304-t004:** Longitudinal analysis of ASES and CMS scores among patients with supraspinatus tears (n = 51).

Score	Examinations			Minimum Detectable Change (MDC)				Variance Analysis	
	Exam 1	Exam 2	Exam 3	ICC (95% CI)	SEM	MDC_95_	Absolute Difference	(*p*/η_p_^2^)	Comparison of Adjacent Exams (η_p_^2^)
**ASES**	45.4 ± 16.4	65.0 ± 16.4	81.1 ± 16.3	0.38 (0–0.65)	16.4	43.8	1.6	**<0.001/0.66**	1 vs. 2 (0.52) 2 vs. 3 (0.58)
**CMS**	41.6 ± 17.5	61.9 ± 17.7	77.5 ± 18.6	0.51 (0.1–0.76)	12.6	42.8	1.2	**<0.001/0.68**	1 vs. 2 (0.61) 2 vs. 3 (0.55)

ICC = intraclass correlation coefficient; SEM = standard error of the mean; ASES = American Shoulder and Elbow Surgeons Score; CMS = Constant–Murley Shoulder Score; Exam 1 = pre-operative; Exam 2 = 3 months post-operative; Exam 3 = 6 months post-operative. Significant and clinically relevant differences are marked in bold.

**Table 5 jpm-13-01304-t005:** Longitudinal analysis of positive IRO/Shift and Jobe tests (n = 51).

Clinical Test	Exam 1	Exam 2	Exam 3	1 vs. 2 Chi²/*p*	2 vs. 3 Chi²/*p*
**IRO/Shift**	92%	61%	10%	2.33/0.127	3.58/0.059
**Jobe**	94%	71%	35%	**7.65/0.006**	**11.59/<0.001**

IRO/Shift test = Internal Rotation and Shift test; Exam 1 = pre-operative; Exam 2 = 3 months post-operative; Exam 3 = 6 months post-operative. Significant relationships marked in bold.

**Table 6 jpm-13-01304-t006:** Differences between single-row, double-row, and debridement repairs compared to ASES and CMS scores.

Score	Surgery	Examinations			Minimum Detectable Change (MDC)				Variance Analysis	
		Exam 1	Exam 2	Exam 3	ICC (95% CI)	SEM	MDC_95_	Absolute Difference	(*p*/η_p_^2^)	Comparison of Adjacent Exams (η_p_^2^)
**ASES**	**Single** (n = 22)	42.1 ± 15.3	64.8 ± 17.3	81.8 ± 14.7	0.33 (0–0.66)	9.19	31.2	10.9	**<0.001/0.72**	1 vs. 2 (**0.62**) 2 vs. 3 (**0.56**)
	**Double** (n = 16)	46.4 ± 15.6	58.9 ± 16.4	73.8 ± 19.0	0.62 (0.07–0.86)	10.5	35.7	10.7	**<0.001/0.62**	1 vs. 2 (**0.51**) 2 vs. 3 (**0.50**)
	**Debride** (n = 13)	49.9 ± 19.0	73.1 ± 11.4	88.9 ± 11.9	0	14.1	47.9	2.0	**<0.001/0.65**	1 vs. 2 (**0.48**) 2 vs. 3 (**0.79**)
**CMS**	**Single** (n = 22)	42.1 ± 16.2	64.3 ± 15.7	78.7 ± 15.0	0.47 (0–0.77)	11.4	38.7	3.4	**<0.001/0.76**	1 vs. 2 (**0.68**) 2 vs. 3 (**0.62**)
	**Double** (n = 16)	38.1 ± 19.9	53.3 ± 16.9	68.8 ± 22.5	0.56 (0.05–0.83)	13.1	44.5	6.4	**<0.001/0.55**	1 vs. 2 (**0.55**) 2 vs. 3 (**0.41**)
	**Debride** (n = 13)	44.8 ± 17.0	68.5 ± 19.0	86.1 ± 15.3	0.38 (0–0.74)	13.5	45.8	1.0	**<0.001/0.73**	1 vs. 2 (**0.62**) 2 vs. 3 (**0.70**)

ICC = intraclass correlation coefficient; SEM = standard error of the mean; ASES = American Shoulder and Elbow Surgeons Score; CMS = Constant–Murley Shoulder Score; Single = single-row surgical repair; Double = double-row surgical repair; Exam 1 = pre-operative; Exam 2 = 3 months post-operative; Exam 3 = 6 months post-operative. Significant differences marked in bold.

**Table 7 jpm-13-01304-t007:** Differences between single-row (n = 22), double-row (n = 16) and debridement (n = 13) repairs compared to IRO/Shift and Jobe tests. Values are the number of positive and negative tests.

Test	Surgery	Exam 1		Exam 2		Exam 3	
		Pos/Neg	% Positive	Pos/Neg	% Positive	Pos/Neg	% Positive
**IRO/Shift**	**Single (n = 22)**	22/0	100	13/9	59	1/21	5
	**Double (n = 16)**	14/2	88	12/4	75	4/12	25
	**Debridement** **(n = 13)**	11/2	85	6/7	46	0/13	0
**Jobe**	**Single (n = 22)**	21/1	95	16/6	73	10/12	45
	**Double (n = 16)**	16/0	100	13/3	81	6/10	38
	**Debridement** **(n = 13)**	11/2	85	6/7	46	2/11	15

IRO/Shift = Internal Rotation and Shift test; Single = single-row surgical repair; Double = double-row surgical repair; Exam 1 = pre-operative; Exam 2 = 3 months post-operative; Exam 3 = 6 months post-operative; Pos = positive test; Neg = negative test.

**Table 8 jpm-13-01304-t008:** Comparison of current study with previous research.

Study	Pre-Operative CMS Score	3-Month Post- Operative CMS Score	6-Month Post- Operative CMS Score	12-Month Post- Operative CMS Score	Time to Return to ADL
Haunschild et al. [31]	--------	--------	--------	--------	8 months
Kukkonen et al. [32]	53	62	--------	76	--------
Nabergoj et al. [33]	54	54	--------	80	--------
Current study	42	62	76	--------	6 months

CMS = Constant–Murley Shoulder Score; ADL = activities of daily living.

## Data Availability

Please contact the corresponding author for inquiries regarding study data.

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
