# Peer review of "Longitudinal Analysis of the ASES and Constant–Murley Scores, and the Internal Rotation/Shift and Jobe Tests Following Arthroscopic Repair of Supraspinatus Lesions"

_jpm, 2023, doi:10.3390/jpm13091304_

Round 1
Reviewer 1 Report
• In the introduction, the difference between the current study and the literature is emphasized.
• Is it possible to create a comparison table in which the results of the studies in the literature and the results of the current study are compared?
• Comparison should be made with the following study: https://doi.org/10.3390/jpm12122018
• References to recent years should be increased (Especially 2023)
Author Response
Response to Reviewer 1 Comments
INTRODUCTION
Point 1: In the introduction, the difference between the current study and the literature is emphasized.
Response 1: Thank you for your positive feedback and the appreciation of our work, especially concerning the introduction.
Point 2: Is it possible to create a comparison table in which the results of the studies in the literature and the results of the current study are compared? Comparison should be made with the following study: https://doi.org/10.3390/jpm12122018.
Response 2: Thank you for this suggestion and we have added a comparison table (table 8); however, the specific suggested study is from our previous work and includes the exact same patient dataset. Therefore, we would be comparing the same patients and patient results, which, in our opinion, would not be appropriate or useful.
Point 3: References to recent years should be increased (Especially 2023).
Response 3: This is a great point and we have performed a new literature research in order to update the references throughout the manuscript.
Reviewer 2 Report
The manuscript "Longitudinal Analysis of the ASES and Constant-Murley Scores, and the Internal Rotation/Shift and Jobe tests following Arthroscopic Repair of Supraspinatus Lesions" is a well planned, designed and documented study. The following comments needs to be addressed before acceptance.
1. The abstract should describe the entire work in a systematic way. Please give conclusive explication in 2-3 lines towards the end of abstract.
2. The introduction is too short and generic. A detailed literature review is missing in the field of relevance. Suggest to include the recent works in the area.
3. Surprised to see that there are no graphical representations made in the entire manuscript. It is very difficulty for the reader to understand the trends and interpret the data. Suggest to represent the data graphically and include the relevant discussion in the manuscript.
4. Consideration of other health conditions of the patients would be meaningful to understand the inter-correlation between the scores and preliminary health complications. Seems like the current articles missed this. If possible try to include atleast some preliminary health conditions.
5. Suggest to combine the results and discussion in to a single section for better technical flow of thought.
6. The conclusion may be rewritten to capture the entire work, preferbly in a bullet point for better readability.
The language quality is good. A minor check is required on sentence construct and mis-spelling.
Author Response
Response to Reviewer 2 Comments
COMMON
Point 1: The manuscript "Longitudinal Analysis of the ASES and Constant-Murley Scores, and the Internal Rotation/Shift and Jobe tests following Arthroscopic Repair of Supraspinatus Lesions" is a well planned, designed and documented study. The following comments needs to be addressed before acceptance.
Response 1: Thank you for your positive feedback and the appreciation of our work.
ABSTRACT
Point 2: The abstract should describe the entire work in a systematic way. Please give conclusive explication in 2-3 lines towards the end of abstract.
Response 2: We attempted to revise the abstract based on this suggestion (see below); however, we were limited due to the word count restriction.
@ line 13-26: It is essential to investigate patients, post-surgery using functional surveys like the American Shoulder and Elbow Surgeons Shoulder (ASES) and Constant-Murley shoulder (CMS) scores, as well as clinical tests such as the Internal Rotation and Shift (IRO/Shift) and Jobe tests. In this study 51 out of an initial 87 patients underwent an arthroscopic supraspinatus repair (22 single-row, 16 double-row, 13 debridement). Testing occurred pre-surgery, and 3- and 6 months post-surgery. Both surveys showed significant improvements over time among all 87 patients, but there were no differences between groups (lesion/no lesion) (p>0.815) or time x group (p>0.895). The IRO/ shift test showed a stronger ability to distinguish between both groups (positive vs. negative) with respect to ASES and CMS scores over time, but the Jobe test did not (p>0.100). Improvements in the CMS scores and Jobe test were lower following repair compared to the ASES and IRO/ shift test. Most patients returned to adequate levels of functional abilities at 6-months post-surgery. The time required to return to activities of daily living and negative clinical tests was longer for the double-row repair patients compared to single-row and debridement. In conclusion, both the functional surveys and clinical tests demonstrated improvements following surgery.
INTRODUCTION
Point 3: The introduction is too short and generic. A detailed literature review is missing in the field of relevance. Suggest to include the recent works in the area.
Response 3: We have conducted a new literature review and provided new and more recent citations and descriptions throughout the introduction section.
Point 4: Surprised to see that there are no graphical representations made in the entire manuscript. It is very difficult for the reader to understand the trends and interpret the data. Suggest to represent the data graphically and include the relevant discussion in the manuscript.
Response 4: In line with your comment, we have added a new flow chart for a better understanding of our participants and study procedures.
@ line 99-101 (Figure 1)
Point 5: Consideration of other health conditions of the patients would be meaningful to understand the inter-correlation between the scores and preliminary health complications. Seems like the current articles missed this. If possible try to include at least some preliminary health conditions.
Response 5: We have added the following statement to try and address your comment.
@ line 84-85: No patient had rheumatic disease; however, information pertaining to patient history of smoking, diabetes mellitus, and thyroid disease was not collected.
Point 6: Suggest to combine the results and discussion in to a single section for better technical flow of thought.
Response 6: Although we agree that combining the results and discussion sections into one section could be helpful, the requirements of the Journal of Personalized Medicine mandate that these must be separated. However, we have revised the discussion section to better include specific results for clarity, while drawing on these results in the discussion.
CONCLUSION
Point 7: The conclusion may be rewritten to capture the entire work, preferbly in a bullet point for better readability.
Response 7: We completely reworked the conclusion as suggested:
@ line 316-327:
ASES and CMS scores, as well as the IRO/ shift and Jobe clinical tests showed significant improvements over time following supraspinatus repair. The main results and suggestions are:
- The IRO/ shift test showed a stronger ability to distinguish between those with positive tests compared to negative tests with respect to ASES and CMS scores.
- The average for all tools did not improve to acceptable levels until approximately 6-months post-surgery, with improvements in the CMS scores and Jobe test being slower at all time periods compared to the ASES and IRO/ shift test.
- The time required to return to activities of daily living and negative clinical tests was longer for the double-row repair patients compared to single-row and debridement.
- These results suggest that the ASES, CMS, IRO/ shift test, and Jobe test are useful, yet unique, tools for determining the progress and clinical decision-making process among patients with supraspinatus repair.
COMMENTS ON THE QUALITY OF ENGLISH LANGUAGE:
Point 8: The language quality is good. A minor check is required on sentence construct and misspelling.
Response 8: We have reviewed the entire manuscript and attempted to correct any grammar and spelling errors.